# Preventing Skeletal-Related Events in Newly Diagnosed Multiple Myeloma

**DOI:** 10.3390/cells14161263

**Published:** 2025-08-15

**Authors:** Benjamin Massat, Patrick Stiff, Fatema Esmail, Estefania Gauto-Mariotti, Patrick Hagen

**Affiliations:** Division of Hematology and Oncology, Department of Medicine, Loyola University Medical Center, Maywood, IL 60153, USA

**Keywords:** multiple myeloma, skeletal-related events, zoledronic acid, denosumab

## Abstract

Despite the increasing number of novel therapies to treat newly diagnosed multiple myeloma (NDMM), preventing skeletal-related events (SREs) remains a challenge. This review summarizes the mechanistic causes of myeloma bone disease, data supporting the use of bisphosphonates and RANKL inhibitors, and the optimal management of preventing SREs in NDMM patients. Both zoledronic acid (ZA) and denosumab are acceptable treatment options with comparable safety and efficacy profiles. However, in patients who are candidates for autologous stem cell transplant (ASCT), denosumab may be preferred over ZA due to a progression-free survival (PFS) benefit observed in post hoc analyses when used with proteasome inhibitor-based regimens. The optimal duration of bone-directed therapy is unclear, but it is typically given for two years. Supportive care should include dental evaluation at baseline, annually, and if symptoms appear, given the risk for jaw osteonecrosis with both ZA and denosumab. Both drugs should be held in the setting of dental work. Patients should receive adequate calcium and vitamin D supplementation. Supportive procedures such as cement augmentation, radiation, and orthopedic surgery can also help treat compression fractures, uncontrolled pain, cord compression, and pathologic fractures. We conclude with our approach for managing SREs and a review of novel therapies and targets.

## 1. Introduction

Multiple myeloma (MM) is a hematologic malignancy that occurs due to the proliferation of malignant monoclonal plasma cells in the bone marrow [1]. An estimated 36,110 new cases will be diagnosed in 2025 in the United States, and the lifetime risk of developing MM is estimated at 0.8% [2]. Skeletal-related events (SREs), defined as a vertebral or non-vertebral fracture, spinal cord compression, radiation or surgery to a bone lesion, and new osteolytic lesions, remain a major source of morbidity impacting health-related quality of life [3]. In a large natural history study, amongst 1027 patients with newly diagnosed multiple myeloma (NDMM) from 1985 to 1998, 67% had lytic lesions, 26% had pathologic fractures, and 22% had compression fractures on conventional radiographs. Further, some had multiple abnormalities, and osteoporosis was present in 23% of patients [4,5]. Despite the increasing number of novel therapies now available and their ability to generate deep, durable responses, preventing SREs remains a challenge due to alterations in bone homeostasis caused by MM cells in the bone marrow microenvironment (in addition to osseous plasmacytomas), leading to increased osteoclast activity and suppressed osteoblast function [4,5].

At the cellular level, normal physiologic bone remodeling involves the resorption of bone by osteoclasts followed by bone formation by osteoblasts [6]. Multiple local and systemic factors help regulate bone remodeling. For example, calcitonin, parathyroid hormone, and vitamin D3 help regulate bone remodeling and blood calcium levels on a systemic level. Additionally, estrogens, androgens, thyroid hormone, and growth hormone influence the bone remodeling systemically. Locally, bone morphogenic proteins (BMPs), transforming growth factor-ß (TGF-ß), epidermal growth factors (EGFs), fibroblast growth factors (FGFs), insulin-like growth factor-1 (IGF-1), and canonical Wnt signaling all modulate bone resorption and formation [6].

In myeloma bone disease, osteoclast activity is upregulated while osteoblast activity is suppressed. To upregulate osteoclast activity, MM cells express receptor activator of nuclear factor-kappa β ligand (RANKL), which binds to RANK on osteoclast precursors, promoting maturation of osteoclasts [5,7]. MM cells also degrade osteoprotegerin (OPG), a soluble decoy receptor for RANKL that antagonizes osteoclastogenesis, after internalizing it via binding to heparan sulfate proteoglycans (HSPGs), such as syndecan-1 (CD138), on the myeloma cell surface [8].

To downregulate osteoblast activity, MM cells express canonical Wnt pathway antagonists such as sclerostin, Dickkopf-related protein 1 (DKK-1), and soluble frizzled-related proteins (sFRPs), which suppress osteoblast activity [9,10]. Semaphorin 4D (Sema4D) is also produced by MM cells and inhibits osteoblastic bone formation [11]. MM cells also express excessive activin A, which pathologically activates TGF-β family receptors on osteoblasts, inhibiting their maturation [12].

In the bone marrow microenvironment, interactions between bone marrow stromal cells, Th17 cells, and MM cells induce cytokine release and secretion of tumor necrosis factor-α (TNF-α), chemokine ligand 3 (CCL3), stromal cell-derived factor-1α (SDF-1α), and annexin A2 (ANXA2), which also promotes osteoclast activity and inhibits osteoblastogenesis [5,13]. MM cells adhere to bone marrow stromal cells via interactions between vascular cell adhesion molecule-1 (VCAM-1) and very late antigen-4 (VLA-4), enhancing osteoclast activity [14,15]. Activation of Notch signaling in osteocytes by myeloma cells induces osteocyte apoptosis, stimulates RANKL expression, and increases Notch receptor expression in myeloma cells, stimulating their growth [16]. These mechanisms are depicted in Figure 1.

Antiresorptive agents such as bisphosphonates and RANKL inhibitors have been shown to both reduce the frequency of and delay time to onset of SREs, and have both been compared in terms of safety and efficacy in several clinical trials as outlined in this review. They may impact overall survival (OS) in MM as well. There is much debate regarding the optimal management of myeloma bone disease and dosing of bone-directed therapy, and physician practices and society guidelines vary widely on the best approaches to prevent SREs in NDMM. The aim of this article is to review available data on the prevention of SREs, outline a practical treatment algorithm for the prevention and management of myeloma bone disease, and review novel therapies and targets.

## 2. Bone-Modifying Agents: Bisphosphonates and RANKL Inhibitors

### 2.1. Bisphosphonates

Table 1 outlines the mechanism of action, route of administration, pharmacokinetics, efficacy in preventing SREs, and impact on survival of both bisphosphonates and RANKL inhibitors. Bisphosphonates bind strongly to hydroxyapatite binding sites on bony surfaces and are incorporated into sites of active bone remodeling and resorption where bone mineral is exposed. This impairs the ability of the osteoclasts to form the ruffled border, thus affecting their ability to initiate bone remodeling, leading to pathologic bone loss in conditions that increase their resorptive activity, like MM. Additionally, bisphosphonates induce osteoclast apoptosis by inhibiting farnesyl pyrophosphate synthase within osteoclasts, a key regulatory enzyme for the production of cholesterol and other lipids [17]. While bisphosphonate molecules that have not been incorporated into the skeleton are cleared rapidly from the plasma, bisphosphonates that have been incorporated into the bone may persist in the bone for a prolonged period, with a half-life potentially greater than 10 years [17,18].

In addition, bisphosphonates were found to have a direct antitumor effect on myeloma cells by inhibiting the release of growth factors from osteoclasts and bone marrow stromal cells. In vitro exposure to high and low concentrations of ZA showed a reduction in both IL-6 production and plasma cell adhesion molecules, thereby reducing proliferation, migration, and attachment of plasma cells in the bone marrow microenvironment. This barrier to myeloma cell binding subsequently reduced the ability for plasma cells to become embedded into the bone marrow stroma and decreased myeloma cell stimulation of osteoclasts [19].

**Table 1 cells-14-01263-t001:** Comparison of agents used for bone-directed therapy in multiple myeloma. Abbreviations: ATP—adenosine triphosphate, CI—confidence interval, CrCl—creatinine clearance, MM—multiple myeloma, OS—overall survival, PFS—progression-free survival, RANKL—receptor activator of nuclear factor kappa-B ligand.

	Mechanism of Action	Route of Administration and Schedule	Pharmacokinetics	Reduction in Skeletal-Related Events	Progression-Free and Overall Survival Benefit
Clodronate	A non-nitrogen-containing (first-generation) bisphosphonate that incorporates into adenosine triphosphate (ATP) after osteoclast-mediated uptake from the bone mineral surface, leading to osteoclast apoptosis [17].	1600 mg by mouth daily indefinitely or until progression of osteolytic lesions or hypercalcemia not responsive to fluids and chemotherapy [20].	Half-life elimination of 4–8 h in animal models Renal elimination Contraindicated if serum creatinine is over 5 mg/dL If CrCl is 30–50, administer 75% of normal oral dose If CrCl is <30, administer 50% of normal oral dose No dose adjustment for hepatic impairment [21]	Significant reduction in progression of osteolytic bone lesions compared to placebo [22].	No statistically significant survival benefit observed with MM-specific mortality. PFS not examined [20].
Pamidronate	A nitrogen-containing (second-generation) bisphosphonate that incorporates into bone and inhibits farnesyl pyrophosphate synthase in osteoclasts after endocytosis, which regulates production of sterols and lipids critical for osteoclast cellular activities, ultimately leading to apoptosis [17].	30 mg of intravenous pamidronate monthly for at least three years [23].	Half-life elimination of 28 +/− 7 h If CrCl < 30 or Cr > 3, consider a longer infusion time of 30 mg over 4–6 h Avoid use and consider alternative agents in dialysis patients Withhold therapy for kidney function deterioration without apparent cause, resume when kidney function returns to within 10% of baseline No dose adjustment for hepatic impairment [22,24]	Significant reduction in skeletal-related events in patients with both newly diagnosed and relapsed MM compared to placebo. This benefit was only observed with intravenous administration, not oral administration [25,26,27].	No statistically significant overall survival benefit observed. PFS not examined [25].
Zoledronic Acid	Same as pamidronate, but 100 times as potent [17]. Additionally, has in vitro direct anti-tumor effects by down-regulating IL-6 secretion and reducing CD40, CD49d, CD54, and CD106 expression [19].	4 mg of intravenous zoledronic acid every four weeks for two years [28].	Half-life elimination of 146 h If CrCl is 50–60, reduce dose to 3.5 mg If CrCl is 40–50, reduce dose to 3.3 mg If CrCl is 30–40, reduce dose to 3 mg Use not recommended for CrCl < 30 or if patient is on dialysis No dose adjustment for hepatic impairment [29,30]	Lower rates of skeletal-related events when compared to clodronate [3].	Reduced mortality by 16% (95% CI 4–26%) and extended median overall survival by 5.5 months (*p* = 0.04) compared to clodronate. Improved progression-free survival compared to clodronate by 12% (95% CI 2–20%) [3].
Denosumab	Human monoclonal antibody that binds and neutralizes RANKL, inhibiting osteoclast maturation and bone resorption [31]. Shown in murine models to reduce serum paraprotein levels [32]. In contrast to bisphosphonates, denosumab is not incorporated into bone and thus its effect on bone resorption rapidly declines upon treatment discontinuation [33,34,35].	120 mg of subcutaneous denosumab every four weeks for two years [36,37].	Half-life elimination is 25.4 days No dosage adjustment necessary for CrCl < 30, monitor closely for hypocalcemia No dose adjustment for hepatic impairment [38]	Denosumab superior regarding time to first skeletal-related event when compared to zoledronic acid after two years of treatment [36].	Overall survival is similar to zoledronic acid, but a progression-free survival benefit was found in patients intended to undergo autologous stem cell transplant (46.1 months vs. 35.7 months, HR 0.65, 95% CI 0.49–0.85, *p* = 0.002) [37].

### 2.2. Clodronate

In the early 1990s, clodronate, a first-generation bisphosphonate with low oral bioavailability and daily oral dosing, was found to inhibit the progression of osteolytic bone lesions when added to chemotherapy for the treatment of MM [9]. Clodronate was also studied in a placebo-controlled trial with the aim of examining OS in patients receiving chemotherapy with or without clodronate. Overall, no OS advantage was found; however, amongst patients with no skeletal fractures at presentation, a significant survival advantage (59 vs. 37 months, *p* = 0.006) was observed. This analysis, however, was no longer statistically significant when confined to myeloma-specific mortality (*p* = 0.22). Rates of SREs were not assessed [20].

### 2.3. Pamidronate

Pamidronate is a second-generation bisphosphonate with a half-life of 28 +/− 7 h [24]. Brickner et al. compared 300 mg daily of oral pamidronate against placebo in patients with NDMM. This randomized multi-center trial found no statistically significant reduction in skeletal-related morbidity, frequency of hypercalcemia, or OS; however, patients treated with pamidronate had fewer episodes of severe pain and a decreased reduction in body height. The overall negative result of the study was thought to be due to low absorption of oral bisphosphonate therapy [26].

A randomized, double-blind study compared 90 mg monthly of IV pamidronate for nine monthly cycles vs. placebo for the reduction in SREs in patients with at least one known skeletal lesion in both NDMM and relapsed disease. The results showed a significant reduction in SREs in the pamidronate group (24% vs. 41%, *p* < 0.001) in both patients receiving first-line or second-line antimyeloma therapy. The proportion of patients with an SRE (28% vs. 59%, *p* = 0.001), pathologic fracture (22% vs. 52%, *p* = 0.006), and radiation treatment to bone (16% vs. 33%, *p* = 0.05) was significantly lower in the pamidronate group compared to placebo. Additionally, time to first SRE (*p* = 0.001 by log-rank test), first pathologic fracture (*p* = 0.006), and first radiation treatment to bone (*p* = 0.05) were significantly shorter in the placebo group. There was no difference in OS or adverse events [27]. Further analysis after an additional 12 cycles of treatment (21 total cycles) revealed a potential survival advantage in patients receiving pamidronate and second-line antimyeloma therapy (14 vs. 21 months); however, OS was not statistically different between the groups (*p* = 0.41) [25].

Gimsing et al. studied the optimal IV dosing of pamidronate by randomizing 504 previously untreated patients with NDMM to receive either 30 mg or 90 mg of IV pamidronate monthly for at least three years. There were no significant differences between the groups when comparing physical function scores, global health quality of life, time to SREs, progression-free survival (PFS), or OS. There was a trend towards fewer incidences of jaw osteonecrosis (ONJ) and treatment discontinuation due to nephrotoxicity in the 30 mg group, though this was not statistically significant. Thus, 30 mg of IV pamidronate was found to be non-inferior to 90 mg dosing [23].

### 2.4. Zoledronic Acid

Zoledronic acid (ZA) is a third-generation bisphosphonate with a half-life of 146 h and a shorter infusion time than pamidronate (15 min vs. two hours) [24,29,30] that may have direct anti-tumor effects. In 2003, Rosen et al. compared ZA to pamidronate treatment for 24 months in patients with Durie–Salmon Stage III MM or breast cancer metastatic to bone. They found that ZA was more effective at reducing the risk of developing skeletal complications in the overall population (risk ratio = 0.841, *p* = 0.030); however, most of this risk reduction was attributable to its higher efficacy in breast cancer patients while maintaining similar efficacy to pamidronate in MM patients. The authors concluded that in MM, ZA had efficacy comparable to that of pamidronate [28].

In 2005, Corso et al. found that ZA, by inducing apoptosis, modifying adhesion molecule expression, down-regulating cytokine secretion, and reducing cell proliferation, showed in vitro antitumor activity against myeloma bone marrow stromal cells and plasma cells. Specifically, it down-regulated IL-6 secretion and reduced CD40, CD49d, CD54, and CD106 expression. The net effect was a dose-dependent decrease in proliferation and an increase in apoptosis of tumor cells [19].

This direct antitumor effect was demonstrated clinically by Morgan et al. with a randomized controlled trial comparing ZA to clodronate in MM. They found that ZA reduced mortality by 16% (95% CI 4–26%), extended median OS by 5.5 months (*p* = 0.04), and improved PFS by 12% (95% CI 2–20%) when compared with clodronate. Response rates and adverse events were similar; however, ZA was associated with higher rates of ONJ (4% vs. < 1%). Rates of treatment discontinuation prior to disease progression were similar in both groups (24% for ZA and 19% for clodronate) [39]. In addition, continued follow-up revealed a lower incidence of SREs in the ZA group [3].

ZA is dosed at 4 mg IV every four weeks [29]; however, longer dosing intervals have been examined as well. For example, the Z-MARK study measured levels of urinary N-telopeptide of type 1 collagen (uNTX) to guide dosing frequency in patients who had received one to two years of prior bisphosphonate therapy. Patients received 4 mg of ZA every 4 weeks if uNTX was >50 nmol/mmol creatinine and every 12 weeks if uNTX was <50 nmol/mmol creatinine. If patients experienced an SRE, had increasing uNTX levels, or had disease progression, they would receive ZA every four weeks. SRE incidence was low, 5.8% during year one and 4.9% during year two, suggesting less frequent dosing may be appropriate in certain patient groups [40]. Himelstein et al. further examined this by randomizing patients with metastatic breast cancer, metastatic prostate cancer, or MM and at least one site of bone involvement to receive ZA every 4 weeks or every 12 weeks for a total of 2 years of therapy. They found that dosing every 12 weeks is non-inferior regarding SREs; however, this dosing interval did not significantly decrease the frequency of adverse events, including ONJ and kidney dysfunction [41].

### 2.5. RANKL Inhibitors

#### Denosumab

Denosumab is a human monoclonal antibody that binds and neutralizes the receptor activator of nuclear factor kappa B ligand (RANKL). RANK (receptor activator of nuclear factor-kappa B) is a surface receptor expressed on osteoclast precursors and is a member of the tumor necrosis factor receptor (TNFR) superfamily. Activation of RANK by RANKL promotes the maturation of osteoclasts, leading to bone remodeling and repair. High-affinity binding of denosumab to RANKL prevents RANKL binding to RANK, thus inhibiting osteoclast maturation and bone resorption. In this way, denosumab mimics the endogenous effects of osteoprotegerin, the decoy receptor for RANKL [31].

Myeloma plasma cell interaction with bone marrow stromal cells is thought to increase secretion of RANKL, therefore stimulating production of activated osteoclasts from monocytic precursors. This leads to osteoclast recruitment of myeloma cells and further promotion of osteoclast proliferation and bone destruction [42]. Pearse et al. have demonstrated that administration of RANK-Fc in a SCID-hu murine model of human myeloma caused a reduction in serum paraprotein as well as a reduction in tumor burden [32], arguing for a direct anti-myeloma effect.

To evaluate denosumab’s clinical anti-myeloma activity, a proof-of-concept phase 2 multicenter single-arm study in patients with relapsed or plateau-phase myeloma investigated whether RANKL inhibition could reduce serum monoclonal protein levels. While no subject met the protocol-defined objective response criteria, denosumab was found to stabilize disease via suppression of the RANKL pathway in 11 (21%) subjects who had relapsed within three months of study enrollment, with a median duration of response of 2.6 months (range of 0.7–16.5 months). A total of 19 (46%) subjects in the plateau-phase cohort maintained stable disease with a median duration of 10.2 months (range of 0.7–18.3 months). These results suggest that denosumab may mediate a direct anti-myeloma effect in some patients. Additionally, the bone turnover marker serum C-terminal telopeptide of type 1 collagen (sCTx) was suppressed in both the plateau-phase and relapsed cohorts, showing inhibition of the RANKL pathway regardless of previous bisphosphonate exposure [43].

In contrast to bisphosphonates, denosumab is not incorporated into bone and thus its effect on bone resorption rapidly declines upon treatment discontinuation. In osteoporosis patients, discontinuation of denosumab results in rising bone turnover markers within three months, a decrease in bone mineral density to baseline levels by 12 months, and an increase in vertebral fractures to a rate similar to untreated populations. This effect, however, has not been studied in MM patients [33,34,35].

## 3. Comparative Trials

Outside of Rosen et al. (2003), as discussed in the previous section [28], no other clinical trials have directly compared pamidronate to ZA or denosumab in NDMM. Multiple trials have compared the efficacy of ZA and denosumab (Table 2). In 2011, Henry et al. compared denosumab with ZA every 4 weeks in patients with advanced cancer and bone metastases (excluding breast and prostate) or MM with at least one lytic lesion. Enrollment in the myeloma stratum was limited to 10% of the study population, and patients were treated for 34 months at the time of the primary analysis. Denosumab was non-inferior to ZA in delaying time to first on-study SRE (hazard ratio = 0.84, 95% CI 0.71–0.98, *p* = 0.0007). Time to first SRE was 20.6 months for denosumab and 16.3 months for ZA, but this difference was not statistically significant when testing for superiority (*p* = 0.06). Additionally, denosumab suppressed urinary N-telopeptide of type 1 collagen (uNTx) to a greater extent than ZA. Regarding adverse events, denosumab led to more frequent hypocalcemia, and ZA led to more frequent renal toxicity. Rates of ONJ were similar in both groups (1.3% for ZA, 1.1% denosumab) [44].

An ad hoc analysis of the above study revealed an OS advantage in the ZA group (HR 2.26, 95% CI 1.13–4.50, *p* = 0.014) [44]. Raje et al. evaluated potential reasons for the favorable survival seen in the subset of MM patients (*n* = 180) in a post hoc analysis. Given that the primary outcome was SREs, the randomization of MM patients did not stratify for myeloma-specific therapeutic or prognostic factors. Consequently, patients in the ZA arm had better performance status, lower ISS stage, and more frequently received proteasome inhibitors and immunomodulatory drugs. In contrast, the denosumab arm had more patients with poor renal function. Other key laboratory and cytogenetic prognostic markers were not collected, and 13 patients in the ZA group withdrew consent during the study. After adjustment with covariate analysis, the CI crossed unity (HR 1.86, 95% CI 0.90–3.84, *p* = 0.0954) [45].

In 2018, Raje et al. compared ZA with denosumab in MM patients with at least one lytic lesion in a randomized controlled trial. Denosumab was found to be non-inferior to ZA for time to first SRE, and median cumulative drug exposure was 15.8 months for denosumab and 14.8 months for ZA. Post hoc landmark analysis at 15 months revealed that denosumab was superior regarding time to first SRE (HR 0.66, CI 0.44–0.98, *p* = 0.039). For this group of patients, the median time on study drug was 25 months for ZA and 24 months for denosumab. OS was similar between groups; however, median PFS favored denosumab (46.1 months for denosumab, 35.4 months for ZA, HR 0.82, CI 0.68–0.99, *p* = 0.036). Toxicity profiles, including rates of ONJ, were similar (4% for denosumab and 3% for ZA). Median time to onset of ONJ was 17.3 months in the denosumab group and 13.6 months in the ZA group, and most had known risk factors. Also of note, 60% of SREs occurred within the first three months of the study, and 81% occurred within the first six months [36].

Terpos et al. further investigated the 10.7-month PFS advantage in the denosumab group of the above study. Ad hoc analyses were performed for patients with intent to undergo autologous stem cell transplant (ASCT), without intent to undergo ASCT, and intent to treat according to age and renal function. The highest PFS benefit favoring denosumab was found in the ASCT intention-to-treat group in patients receiving triplet regimens such as VCD, RVD, and VTD, as well as the doublet regimen VD. The PFS benefit was not limited to those who later received an ASCT. No PFS benefit was seen in the group without intent to undergo ASCT. Additionally, a PFS benefit was observed in patients with a CrCl over 60 and in those younger than 70 years old [37].

These results suggest a synergistic effect between bortezomib and denosumab that could help explain the PFS benefit. Bortezomib has been shown to reduce circulating RANKL and DKK-1 levels in MM [46]. Additionally, proteosome inhibitors reduce osteoclast differentiation and stimulate bone formation [47,48,49,50]. It appears, however, that immunomodulatory drugs such as lenalidomide and thalidomide, commonly administered concurrently with proteosome inhibitors, inhibit osteoblast development in vitro, which could negate the positive effects bortezomib has on bone formation [51]. Notably, the RVD group alone did not demonstrate a PFS benefit when administered with denosumab. Overall, denosumab’s PFS benefit may be derived from inhibition of osteoclast activity and subsequent decreased stimulation of tumor cell growth [37].

This PFS advantage for denosumab was investigated further by Mohyuddin et al. in a retrospective cohort study including 1725 denosumab-treated patients and 1515 bisphosphonate-treated patients. Their group found no statistically significant PFS or OS difference in propensity score-weighted models between the groups [52]. However, while patients were stratified by high-risk cytogenetic abnormalities, they were not stratified by ISS or R-ISS stages, and patients in the denosumab group had significantly higher rates of renal dysfunction (defined as GFR < 60). These factors can adversely affect an individual’s risk of relapse and survival. Lastly, the PFS benefit for denosumab was found in patients with the intent to undergo ASCT, those with CrCl over 60, and those younger than 70. Given that few patients went on to receive ASCT in this study (3.8% in the bisphosphonate group and 2.8% in the denosumab group) and patients in the denosumab group were older with higher rates of renal dysfunction, we do not feel that this cohort of patients is representative of the group that had a PFS benefit when receiving denosumab. We interpret these findings as supporting the non-inferiority of denosumab relative to ZA. Although the retrospective cohort by Mohyuddin et al. did not demonstrate a significant PFS or OS benefit, the study population differed substantially from those included in prior prospective trials. Therefore, while the data do not confirm a PFS benefit, they also do not refute the possibility of such a benefit in appropriately selected transplant-eligible patients receiving proteasome inhibitors.

## 4. Supportive Care and Management of Adverse Events

### 4.1. Exercise, Weight Loss, and Lifestyle Modifications

Lifestyle modifications, including maintaining sufficient calcium and vitamin D levels, weight-bearing or resistance exercise, smoking cessation, and limiting alcohol intake, are essential for good bone health [53]. Regular weight-bearing physical activity improves bone health through activation of osteocyte signaling and anabolic effects on bone mineral content and bone mineral density [54]. In the postmenopausal setting, moderate exercise (walking on a treadmill at 50% of maximum heart rate for 30 min, 3 times a week) can reduce bone resorption and turnover markers in as little as one month [55]. While structured exercise programs in MM patients have not been shown to improve objective measures of physical function, they are safe [56], and regular exercise is likely beneficial to MM patients who can tolerate it. To our knowledge, no specific data exists regarding bone resorption and turnover in the setting of exercise in MM patients.

### 4.2. Calcium and Vitamin D Supplementation and Hypocalcemia

Vitamin D is essential for intestinal calcium absorption and maintaining bone health. Adequate calcium and vitamin D intake is especially important when using bone-modifying agents, given the risk of hypocalcemia with treatment. Pre-existing hypocalcemia needs to be corrected prior to starting bone-modifying agents, and, if symptomatic hypocalcemia develops, bone bone-modifying agent should be held until normal levels are restored. The recommended daily intake for calcium is 1200 mg per day for women older than 50 and men older than 70, and 1000 mg for men aged 51–70. For men and women ages 51–70, the recommended vitamin D intake is 600 IU for ages 51–70 and 800 IU after the age of 70 [57,58]. Though monitoring vitamin D levels is generally unnecessary in the general population due to its wide therapeutic index, it may be appropriate to recheck serum 25 (OH)D levels in MM patients who present with severe deficiency below 30 ng/mL (75 nmol/L) after 8–12 weeks. It can also be reassessed after six months in patients with concurrent renal failure, liver failure, metabolic bone diseases, malabsorption, obesity, hypogonadism, or prolonged glucocorticoid treatment at risk for persistent hypovitaminosis D, as well as those who are at risk of hypercalcemia due to an underlying disease where 1,25 (OH)_2_D is appropriate for monitoring such as sarcoidosis [59].

### 4.3. Jaw Osteonecrosis

ONJ is defined as the presence of exposed bone in the maxillofacial region that does not heal within eight weeks after identification by a health care provider in patients treated with bisphosphonates who did not receive radiation therapy to the craniofacial region. Typically, soft tissue closure after trauma, dental extractions, and oral surgical procedures occurs within eight weeks [60]. ONJ is divided into four stages. Stage 0 patients have no clinical evidence of necrotic bone but present with nonspecific symptoms, increased tooth mobility, periapical fistula, or radiographic changes. About 50% of stage 0 patients progress to a higher stage, but patients with stage 0 disease are not considered to have ONJ. Stage 1 patients have exposed and necrotic bone or a fistula but are asymptomatic and have no evidence of infection. Stage 2 patients additionally have pain and evidence of infection. Lastly, stage 3 patients in addition have at least one of the following features: exposed necrotic bone extending beyond the alveolar bone, pathologic fracture, extraoral fistula, oro-antral communication, or osteolysis [61].

The SWOG S0702 prospective observational cohort assessed the incidence and risk factors related to the development of ONJ in various cancer patients receiving ZA. The study found a 3-year incidence rate of ONJ to be highest in patients with MM at 4.3%, a rate higher than lung, prostate, and breast cancer. Additional risk factors for the development of ONJ included higher ZA exposure, fewer teeth, dentures, prior oral surgery, and current smoking [62]. In their 2018 study comparing denosumab with ZA in the treatment of bone disease in MM patients, Raje et al. reported a similar incidence of ONJ between the denosumab and ZA groups (4% vs. 3%; *p* = 0.147). The median time to onset of ONJ was 17.3 months for denosumab and 13.6 months for ZA [36].

Regarding management of ONJ, conservative therapy with optimization of oral hygiene, chlorhexidine mouth rinses, and systemic antibiotics if infection is present can be recommended. Bone-directed therapy can be continued with stage 0 and 1 disease, but discontinuation should be considered at stage 2. Surgical management can additionally be considered at stage 2 [63]. In our practice, we hold all bone-directed therapy when there is any concern for ONJ until x-rays are performed and patients have a full dental evaluation, particularly if patients are beyond 6 months of therapy, given 80% of SREs occur during this time.

### 4.4. Patients with Baseline Renal Impairment

In patients with baseline renal impairment, denosumab can be used at any level of renal function, or the dose of ZA can be reduced if creatinine clearance (CrCl) allows. For patients with a CrCl of 50–60 mL/min, 40–49 mL/min, and 30–39 mL/min, 3.5, 3.3 mg, and 3 mg doses of ZA are recommended, respectively, compared to 4 mg doses recommended with CrCl > 60 mL/min as outlined in Table 1. Additionally, few patients receiving ZA with a creatinine greater than 2 mg/dL have been studied, arguing for an alternate agent for this patient population. Pamidronate infusion can be slowed to be given over 4–6 h for patients with CrCl < 30 or serum creatinine >3 mg/dL, as 30 mg doses are non-inferior to 90 mg doses even in patients with normal renal function.

Renal function should be monitored closely in patients receiving ZA and pamidronate, and doses should be withheld for acute renal deterioration, defined as a rise in creatinine of 0.5 mg/dL in patients with normal baseline creatinine and a rise in creatinine of 1.0 mg/dL in patients with abnormal baseline creatinine. Doses should also be held for low serum calcium levels. Regarding denosumab, patients with baseline renal impairment or on hemodialysis are at increased risk of developing severe symptomatic hypocalcemia. Calcium should be monitored closely and corrected prior to doses [23,24,29,30,38,58].

## 5. Recommendations for the Management of Bone Disease in Newly Diagnosed Multiple Myeloma

Table 3 outlines the current IMWG, NCCN, and ASCO guidelines on the management of bone disease in NDMM patients. Our practice is outlined here and depicted in Figure 2. All patients should be started on bone-directed therapy as soon as possible after MM diagnosis, as SREs often occur within the first few months after diagnosis. From our perspective, both ZA and denosumab are acceptable treatment options to prevent SRE in NDMM patients. ASCT remains the standard of care for transplant-eligible patients based on the FORTE, DETERMINATION, and IFM 2009 trials [64,65,66], and denosumab is likely preferred over ZA in patients with plans to undergo ASCT, given the PFS benefit when combined with proteosome inhibitors in this patient population [37]. Additionally, on a post hoc landmark analysis starting at 15 months, denosumab was superior regarding time to first SRE when compared with ZA (HR 0.66, CI 0.44-0.98, *p* = 0.039) [36] and can be given in cases of renal impairment. However, it has insufficient data regarding treatment discontinuation in MM patients and is not incorporated into the bone. Given that in osteoporosis patients, discontinuation of denosumab results in rising bone turnover markers within 3 months, decrease in bone mineral density to baseline levels by 12 months, and an increase in vertebral fractures to a rate similar to untreated populations [33,34,35], our practice is to start either maintenance therapy with denosumab every 6 months after treatment discontinuation or give a consolidation bisphosphonate dose 6 months after discontinuation. More data is needed to definitively guide the best practices of denosumab discontinuation in MM patients.

The optimal duration of bone-directed therapy is unclear. Dosing with ZA should be given monthly for at least the first year and up to two years. It is possible that with deeper remissions achieved with novel quadruplet-induction therapies, prolonged use of bone-directed therapy exposes patients to unnecessary adverse events. Terpos et al. (2021) recommend considering discontinuing or reducing the frequency of ZA at one year if patients have at least a very good partial response, as there was no benefit found in extending therapy with ZA from two to four years [58,69]. However, insufficient data exists regarding discontinuation of denosumab. When compared head-to-head with ZA by Raje et al., patients were treated with denosumab for approximately two years [36].

Finally, although cost is not the focus of review, on review of data available at our institution wholesale cost (manufacturer price) of denosumab is USD 3285.47, whereas ZA is USD 44.22. However, cost-effectiveness analyses performed by Raje et al. (2018) in the United States of America and Terpos et al. (2019) in Austria, Belgium, Greece, and Italy showed denosumab to be cost effective versus ZA, thought to be due to its strong ability to prevent SREs, possible PFS benefit in transplant-eligible patients receiving proteosome inhibitors, and lack of renal toxicity [70,71]. Regardless, in resource-limited settings, the benefits from the lower cost of ZA may outweigh the potential advantages of denosumab.

Regarding the optimal approach to bone-directed therapy in relapsed disease, there is limited data. In 2015, Garcia-Sanz et al. randomized patients to receive monthly ZA for one year or supportive care at asymptomatic biochemical relapse and found that there was a lower incidence of SREs; however, sample sizes were small, there was no difference in time to next therapy, and there was no significant difference in OS [72]. Despite this, given the likely significant benefit regarding reduction in SREs, we do recommend restarting bone-directed therapy with ZA or denosumab at relapse for at least one year.

Regarding supportive care, we agree with recommendations from Terpos et al. (2021) [58]. In summary, patients should receive a dental evaluation at baseline and annually or if symptoms appear. Bisphosphonates and denosumab should be held in the setting of dental work. Patients should receive adequate calcium and vitamin D supplementation as detailed above. Supportive procedures such as cement augmentation, radiation, and orthopedic surgery can also be considered to help treat compression fractures, uncontrolled pain, cord compression, and pathologic fractures.

## 6. Novel Therapies

Romosozumab, a sclerostin inhibitor FDA-approved for the treatment of osteoporosis in postmenopausal women [73], has potential efficacy in MM bone disease. Sclerostin is a canonical Wnt-pathway antagonist that inhibits bone formation by osteoblasts. The canonical Wnt pathway controls cell proliferation by activating target genes via the transcription factors ß-catenin-T-cell factor and lymphoid enhancer-binding factor. Notably, this pathway helps regulate bone metabolism by influencing the proliferation and differentiation of mesenchymal stem cells and osteoblast progenitor cells. Activation of it also helps regulate bone resorption by osteoclasts [74]. Elevated sclerostin levels result in increased bone resorption and suppressed bone formation [75]. Additionally, sclerostin levels are known to be elevated in MM patients, and higher levels are negatively correlated with survival [76]. Further, deletion of the gene that encodes sclerostin, *Sost*, in a MM mouse model decreased both the number and area of osteolytic lesions compared to *Sost* wild-type mice [9]. Injections of a sclerostin monoclonal antibody also reduced the number of osteolytic lesions in a MM mouse model compared to IgG injections [9]. Anti-sclerostin antibody and ZA together increased bone mass and fracture resistance compared with ZA alone in a mouse model, but these parameters did not differ from anti-sclerostin antibody alone [77]. These data show the potential efficacy of sclerostin inhibition with romosozumab in the treatment of myeloma bone disease, and it is currently being investigated in a phase I trial in postmenopausal women with osteoporosis and MM (NCT05775094). Regarding safety, the ARCH and BRIDGE trials noted that romosozumab slightly increases the risk of adverse cardiovascular events and thus, it is recommended that the drug not be initiated in patients who have had cardiovascular events in the past year and be used with caution in patients with cardiovascular risk factors [73,78,79].

Another potential therapeutic target for myeloma bone disease in the canonical Wnt/ß-catenin pathway is Dickkopf-related protein 1 (DKK-1), an inhibitor of the pathway secreted by myeloma cells in addition to osteoblasts and bone marrow stromal cells (BMSC). It inhibits osteoblast maturation, leading to decreased bone mineral density. High DKK-1 levels have also been positively correlated with higher ISS stage and the presence of increasing numbers of bone lesions [80]. Multiple mechanisms have been examined to therapeutically inhibit DKK1 activity, and BHQ880, a phage-derived DKK1 antibody, has shown potential efficacy in myeloma bone disease [81]. Fulciniti et al. showed that BHQ880 increased osteoblast differentiation and inhibited MM cell growth in the presence of BMSCs in vitro. Additionally, they showed BHQ880 led to inhibition of MM cell growth, increased osteoblast number, and increased trabecular bone in a murine model [82]. It has also been studied in multiple early-phase clinical trials in combination with ZA in relapsed/refractory MM [83,84] and as a single agent in patients with smoldering multiple myeloma [85]. It was well tolerated in all three trials and, when used as a single agent, led to a statistically significant increase in vertebral strength by quantitative computed tomography (qCT) with finite element analysis. However, there were no significant changes in dual-energy x-ray absorptiometry (DXA) measurements, and no direct antimyeloma effects were observed [85]. Further prospective trials, including those in NDMM patients, will be needed to establish BHQ880′s efficacy in reducing SREs.

The transforming growth factor-beta (TGF-β) signaling pathway is an important regulator of the bone marrow microenvironment and bone remodeling [12,86]. In MM, excessive activin A expression pathologically activates the TGF-β family receptors on osteoblasts, inhibiting their maturation [12]. Increased expression of activin A has been correlated with osteolytic disease in MM patients, and inhibition of it reversed myeloma-induced osteoblast inhibition and impaired tumor growth in a mouse model [87]. Sotatercept is an activin signaling inhibitor that has a high affinity for multiple TGF-β ligands, aiming to restore the balance of the TGF-β signaling pathway branches [12]. It is currently FDA-approved to treat pulmonary hypertension [88]. Sotatercept was evaluated in a phase IIa trial in MM patients with osteolytic lesions by Abdulkadyrov et al. in 2014 [89]. Patients received four 28-day cycles of escalating doses of sotatercept or placebo with six cycles of melphalan, prednisolone, and thalidomide. Patients receiving sotatercept tolerated it well and had improvements in bone mineral density and bone formation relative to placebo in addition to increases in hemoglobin levels [89]. Further prospective trials are needed to elucidate the role of sotatercept in the treatment of myeloma-related bone disease, and patients with concurrent anemia may benefit from this drug if it is found to reduce rates of SREs with similar efficacy to bisphosphonates or denosumab. The mechanisms of these drugs are depicted in Figure 3.

## 7. Conclusions

Both ZA and denosumab are strong options to prevent SREs in MM, with denosumab likely preferred in patients with plans to undergo ASCT and receiving a proteosome inhibitor. Supportive measures, including dental care, calcium supplementation, and vitamin D supplementation, are paramount, while radiation therapy, cement augmentation, and surgery can be considered for symptomatic lesions. In future studies, shorter durations of bone-directed therapy in patients with strong treatment responses in the era of quadruplet-induction regimens should be examined prospectively. Additionally, different methods of discontinuing denosumab should be assessed to determine the need for maintenance denosumab or a consolidative dose of a bisphosphonate. The PFS benefit seen in denosumab-treated patients should be confirmed with further prospective trials.

## Figures and Tables

**Figure 1 cells-14-01263-f001:**
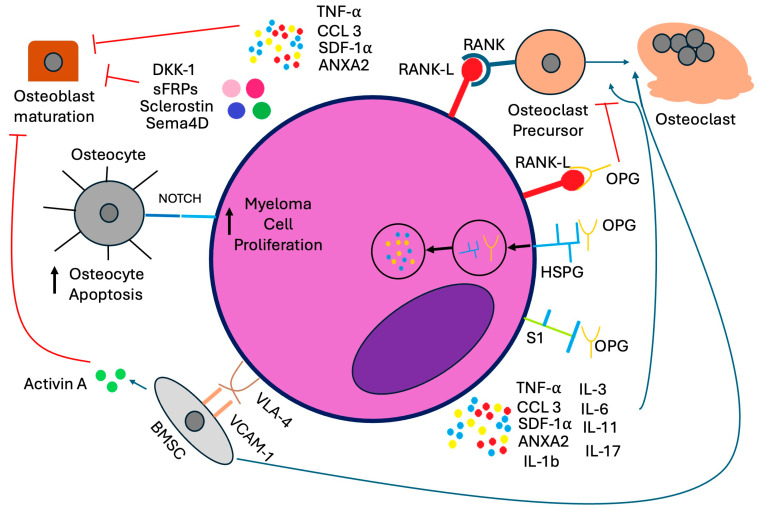
Cellular mechanisms of myeloma bone disease. Abbreviations: ANXA2: annexin A2, BMSC: bone marrow stromal cell, CCL3: chemokine ligand 3, DKK-1: Dickkopf-related protein 1, HSPG: heparan sulfate proteoglycan, OPG: osteoprotegerin, RANKL: receptor activator of nuclear factor-kappa β ligand, S1: syndecan-1, SDF-1α: stromal cell-derived factor-1α, Sema4D: Semaphorin 4D, sFRPs: soluble frizzled-related proteins, TNF-α: tumor necrosis factor-α, VCAM-1: vascular cell adhesion molecule-1, VLA-4: very late antigen-4.

**Figure 2 cells-14-01263-f002:**
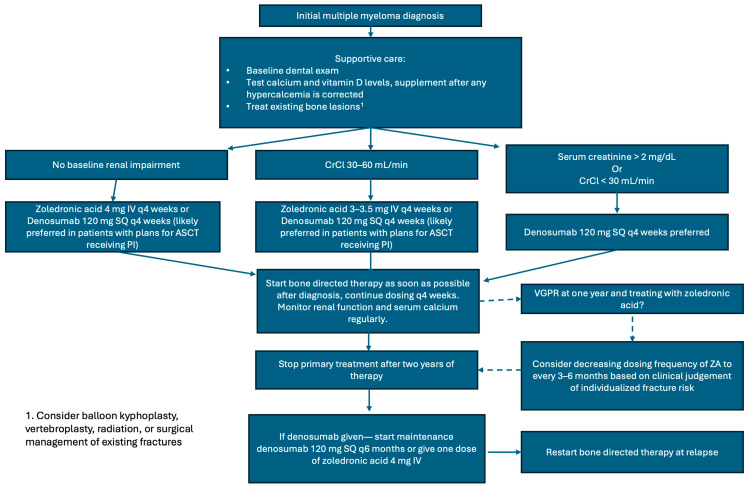
Treatment algorithm for bone-directed therapy in newly diagnosed multiple myeloma. Abbreviations: ASCT: autologous stem cell transplant; PI: proteosome inhibitor; VGPR: very good partial response; ZA: zoledronic acid.

**Figure 3 cells-14-01263-f003:**
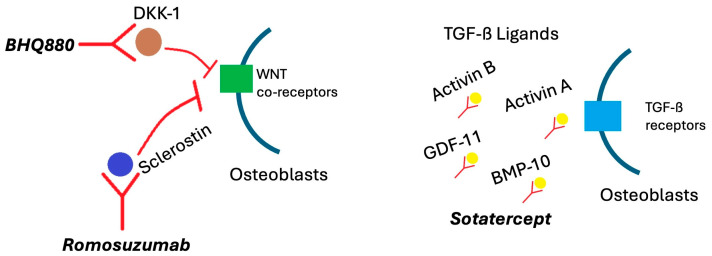
Cellular mechanisms of romosuzumab, sotatercept, and BHQ880 in multiple myeloma.

**Table 2 cells-14-01263-t002:** Summary of trials comparing denosumab and zoledronic acid.

Trial	Drug	Median Time to First Skeletal-Related Event	Median PFS	Median Overall Survival	Grade ≥ 3 Adverse Events Related to Study Drug	Adverse Events Associated With Renal Toxicity	Hypocalcemia Incidence	Jaw Osteonecrosis Incidence
Henry ^1^ [44].	Denosumab	20.6 months ^2^	Not reported, approximately 9 months ^3^	Not reported, approximately 12 months ^4^	Not reported, total AEs leading to treatment discontinuation 10% (*p* = 0.20)	8.3%	10.8%	1.1%
	Zoledronic Acid	16.3 months ^2^	Not reported, approximately 9 months ^3^	Not reported, approximately 12 months ^4^	Not reported, total AEs leading to treatment discontinuation 12% (*p* = 0.20)	10.9%	5.8%	1.3%
Raje/Terpos ^5^ [36,37].	Denosumab	22.8 months ^6^	46.1 months ^7,8^	49.5 months ^9^	5%	10%	17%	4%
	Zoledronic Acid	24 months ^6^	35.7 months ^7,8^	49.5 months ^9^	6%	17%	12%	3%

^1^ Key inclusion criteria: ≥18 years old with solid tumor or MM and at least one bone metastasis or lytic bone lesion, CrCl ≥ 30, and ECOG PS ≤ 2. Key exclusion criteria: breast or prostate cancer, prior treatment with IV bisphosphonate, planned radiation or surgery to bone, or unhealed dental/oral surgery. Approximately 10% of patients in the study had MM. Primary analysis was conducted 34 months after enrollment initiated [44]; ^2^ HR 0.84; 95% CI 0.71 to 0.98; *p* = 0.0007 for non-inferior, 0.06 for superiority [44]; ^3^ HR 1.00, 95% CI 0.89 to 1.12, *p* = 1.0 [44]; ^4^ HR 0.95, 95% CI 0.83 to 1.08, *p* = 0.43 [44]. ^5^ Key inclusion criteria: ≥18 years old with at least one documented lytic bone lesion, adequate performance status, and adequate organ function with CrCl > 30. Key exclusion criteria: plasma cell leukemia, >1 dose of IV bisphosphonate, and non-healed dental/oral surgery. Treatment was planned until approximately 676 patients had an on-study SRE, and the primary efficacy and safety analysis was completed. ^6^ Post hoc analysis after a median of 25 months of ZA treatment and 24 months of denosumab treatment revealed that denosumab was superior regarding time to first SRE (HR 0.66, CI 0.44–0.98, *p* = 0.039). A total of 60% of all first skeletal-related events occurred within the first three months of the study, and 80% occurred within the first six months [36]; ^7^ HR 0.65, CI 0.45–0.85, *p* = 0.002 [37]. ^8^ Ad hoc analysis showed this progression-free survival benefit was found specifically in patients who had intent to undergo autologous stem cell transplant [37]; ^9^ HR 0.90, CI 0.70–1.16, *p* = 0.41 [36].

**Table 3 cells-14-01263-t003:** Summary of available guidelines for bone-directed therapy in multiple myeloma.

	Preferred Agent	Frequency of Treatment and Dosing	Duration of Treatment	Modify Duration Based on Response	Supportive Care	Notes
IMWG: Terpos et al.; Lancet Oncology 2021 [58]	Zoledronic acid (ZA) preferred If patient has no evidence of bone disease, ZA preferred If patient has renal impairment, denosumab preferred	4 mg monthly intravenously for ZA 120 mg monthly for denosumab	At least 12 months If denosumab is discontinued, a single dose of ZA should be given at least six months after discontinuation of denosumab, or denosumab should continue to be administered every six months	If patients achieve ≥ VGPR, can decrease ZA dosing frequency to every 3 months, 6 months, yearly, or discontinue treatment Denosumab treatment should be continued until unacceptable toxicity occurs, but can be discontinued after 24 months, and if the patient achieves ≥ VGPR	Dental evaluation for all patients at baseline and annually, or if symptoms appear Calcium and vitamin D supplementation Cement augmentation can help treat painful vertebral compression fractures. Radiation can help uncontrolled pain, symptoms related to cord compression, and pathologic fractures. Surgery is indicated for prevention or treatment of long bone pathologic fractures, vertebral column instability, or spinal cord compression	Denosumab may prolong progression-free survival in NDMM with related bone disease in patients who are eligible for autologous transplant, but discontinuation is challenging due to the rebound effect.
NCCN: Kumar et al.; JNCCN 2020 [67]	Bisphosphonate [category 1] or denosumab preferred. Denosumab is preferred in patients with renal insufficiency	Frequency of dosing (monthly vs. every three months) depends on the patient, response to therapy, and agent used	Up to two years	Continuing beyond two years is based on clinical judgment	A baseline dental exam is recommended Assess vitamin D status Monitor for renal dysfunction with use of bisphosphonates Monitor for osteonecrosis of the jaw	Patients who discontinue denosumab should be given maintenance denosumab every six months or a single dose of bisphosphonate
ASCO Guidelines [68]	Pamidronate or zoledronic acid preferred with denosumab as a non-inferior alternative preferred in patients with renal insufficiency	90 mg IV pamidronate or 4 mg IV zoledronic acid every 3–4 weeks. Consider reducing the pamidronate dose in cases of renal impairment.	Up to two years	Less frequent dosing should be considered for patient with responsive or stable disease. Continuous use is at the discretion of the treating physician.	Comprehensive dental exam before starting bone-modifying therapy is recommended Calcium and vitamin D should be repleted Monitor for renal dysfunction during treatment Evaluate for albuminuria every 3–6 months for patients on bisphosphonates. Consider discontinuing the drug in patients who develop unexplained urine albumin of >500 mg/24 h.	Retreatment should be initiated at the time of disease relapse

Abbreviations: IMWG: International Myeloma Working Group, NCCN: National Comprehensive Cancer Network, ASCO: American Society of Clinical Oncology, VGPR: very good partial response.

## Data Availability

No new data were created or analyzed in this study.

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
