# Peer review of "Preventing Skeletal-Related Events in Newly Diagnosed Multiple Myeloma"

_cells, 2025, doi:10.3390/cells14161263_

Round 1
Reviewer 1 Report
Comments and Suggestions for Authors
The study discusses the causes of myeloma bone disease, the use of bisphosphonates and RANKL inhibitors, and the optimal management of preventing skeletal related events in newly diagnosed multiple myeloma (NDMM) patients. Both zoledronic acid (ZA) and denosumab are acceptable treatment options, with denosumab likely preferred due to progression-free survival benefits. The optimal duration of bone-directed therapy is typically two years, with supportive care including dental evaluation, calcium and vitamin D supplementation, and supportive procedures. There are several issues that must be addressed prior to considering publication.
- Lack of Clear Novelty or Thesis Statement: While the manuscript is comprehensive, it lacks a clearly stated central argument or novel contribution. This is especially important for a review article. It summarizes available data well but doesn't clearly articulate what is new, what the key controversies are, or how this review adds value beyond existing reviews.
- Overreliance on Secondary and Post Hoc Analyses: Certain recommendations—like the preference for denosumab instead of ZA—are primarily founded on post hoc or subgroup analyses (for instance, the Raje/Terpos trials). The presentation of these should be approached with greater caution, ensuring a clearer differentiation between findings that generate hypotheses and those that constitute robust evidence.
- Inconsistent Use of Data to Justify Recommendations: The manuscript recommends denosumab based on PFS advantage in a subgroup of transplant-eligible patients, but later acknowledges that real-world studies did not replicate this benefit. The conclusion should better reconcile conflicting evidence and be more nuanced about patient selection.
- Missing Discussion on Cost-Effectiveness in Clinical Decision-Making: Although a paragraph briefly mentions cost, the real-world implications of cost vs. benefit (e.g., $3285 vs. $44 per dose) are not fully explored. In resource-limited settings, cost may outweigh marginal PFS benefits.
- The terms NDMM, MM, newly diagnosed multiple myeloma are used interchangeably. Recommend consistent usage of abbreviations (e.g., define NDMM once and use it throughout).
- Insufficient Discussion on Denosumab Discontinuation Risks: While the rebound risk is briefly mentioned, the clinical implications of denosumab discontinuation (e.g., risk of vertebral fractures) should be explored in more depth, especially since it's a key difference from bisphosphonates.
The clarity of the English could be enhanced to better convey the research.
Author Response
Lack of Clear Novelty or Thesis Statement: While the manuscript is comprehensive, it lacks a clearly stated central argument or novel contribution. This is especially important for a review article. It summarizes available data well but doesn't clearly articulate what is new, what the key controversies are, or how this review adds value beyond existing reviews.
Thank you for this comment. We have adjusted our introduction to better reflect controversies in the field and how our review adds value (varying treatment practices and guidelines amongst physicians and societies, a new practical treatment algorithm, and a review of novel therapies/targets). (Lines 94-100)
Overreliance on Secondary and Post Hoc Analyses: Certain recommendations—like the preference for denosumab instead of ZA—are primarily founded on post hoc or subgroup analyses (for instance, the Raje/Terpos trials). The presentation of these should be approached with greater caution, ensuring a clearer differentiation between findings that generate hypotheses and those that constitute robust evidence.
Thank you for pointing this out. We agree that the language can be clearer and approached with greater caution. We have adjusted the manuscript to state that denosumab is likely preferred in patients with plans to undergo ASCT and receiving proteosome inhibitor treatment. These changes were made to the abstract, Recommendations for the Management of Bone Disease in NDMM section, and conclusion. Figure 2 was also adjusted to reflect these changes.
Inconsistent Use of Data to Justify Recommendations: The manuscript recommends denosumab based on PFS advantage in a subgroup of transplant-eligible patients, but later acknowledges that real-world studies did not replicate this benefit. The conclusion should better reconcile conflicting evidence and be more nuanced about patient selection.
We appreciate this comment. We believe you are referring to the paragraph discussing the Mohyuddin et al (2024) retrospective cohort discussed on lines 449-465. While we acknowledge this real-world study did not replicate the PFS benefit, we do not feel that it refutes the PFS advantage in the transplant eligible subgroup noted by Terpos, namely due to the small number of transplant eligible patients in the study (2.8% and 3.8% in each treatment group) among other reasons discussed. However, your point that the conclusion should reflect a more nuanced patient selection for this benefit is well taken. We have adjusted the manuscript to state that denosumab is likely preferred in patients with plans to undergo ASCT and receiving proteosome inhibitor treatment to make this distinction more clear throughout the manuscript.
Missing Discussion on Cost-Effectiveness in Clinical Decision-Making: Although a paragraph briefly mentions cost, the real-world implications of cost vs. benefit (e.g., $3285 vs. $44 per dose) are not fully explored. In resource-limited settings, cost may outweigh marginal PFS benefits.
Thank you for this comment. We agree that this review does not fully explore the cost benefit implications. The cited papers in this paragraph (lines 624-632) do so on a more detailed level. We also agree that in resource-limited settings, the lower cost of ZA may outweigh potential benefits of denosumab, and have added a statement reflecting this. (lines 630-631). Additionally, we are open to omitting this paragraph if the journal feels that cost discussion is not relevant to this review. We did feel this was worth at least briefly covering given the rising costs of myeloma care overall.
The terms NDMM, MM, newly diagnosed multiple myeloma are used interchangeably. Recommend consistent usage of abbreviations (e.g., define NDMM once and use it throughout).
Thank you for pointing this out. We have adjusted the manuscript to make more consistent use of these abbreviations. We have used MM to describe the disease entity of multiple myeloma and NDMM to describe patients with newly diagnosed multiple myeloma.
Insufficient Discussion on Denosumab Discontinuation Risks: While the rebound risk is briefly mentioned, the clinical implications of denosumab discontinuation (e.g., risk of vertebral fractures) should be explored in more depth, especially since it's a key difference from bisphosphonates.
Thank you for this comment and we agree with your point. To our knowledge, this discontinuation effect has only been studied in osteoporosis patients. Our paragraph on recommendations for the management of bone disease in NDMM has been adjusted to make it clearer that our recommendations are based off osteoporosis data and that more data is needed to guide best practices of denosumab discontinuation in NDMM patients. Lines 588-596
Reviewer 2 Report
Comments and Suggestions for Authors
The authors provide a review of the mechanistic causes of bone disease in multiple myeloma, as well as an overview of available drugs or drugs in development to counteract bone loss.
The review is well-organized and informative, and I only have a few minor comments.
Line 141: I am confused by this sentence. I would expect that "Time to the first SRE" would refer to lengths of times instead of percentages such as "28% vs 59%." Do the percentages instead refer to the percent of patients who experienced an SRE by a certain length of time? Similarly, the time to first SRE being "lower in the pamidronate group" (line 143) does not seem correct.
Line 144: A p value of 0.41 is quite high and would not really suggest anything to me.
Line 220: I'm a little unclear on what "both cohorts" refers to. Was there a relapsed cohort and a plateau-phase cohort?
Lines 261 and 285: Do you mean "post hoc" instead of "ad hoc"?
References: I am not a fan of citing UptoDate; it may change over time and have some element of opinion. I would prefer citing references found within UptoDate, or a package insert that includes its year of publication.
Author Response
Line 141: I am confused by this sentence. I would expect that "Time to the first SRE" would refer to lengths of times instead of percentages such as "28% vs 59%." Do the percentages instead refer to the percent of patients who experienced an SRE by a certain length of time? Similarly, the time to first SRE being "lower in the pamidronate group" (line 143) does not seem correct.
Thank you for pointing this out. The percentages noted are proportions of patients with SRE, pathologic fracture, and radiation treatment to bone in the pamidronate and placebo groups. Time to SRE was noted to be shorter in the placebo group as well. These changes are reflected from lines 234-239.
Line 144: A p value of 0.41 is quite high and would not really suggest anything to me.
Thank you for this comment. This line is meant to address the seven-month difference in survival between the two groups in the study. We have moved the p-value to be noted after our comment that the difference between groups is not statistically significant to help avoid confusion (line 253)
Line 220: I'm a little unclear on what "both cohorts" refers to. Was there a relapsed cohort and a plateau-phase cohort?
That is correct. The line has been changed to reflect this more clearly. Line 340
Lines 261 and 285: Do you mean "post hoc" instead of "ad hoc"?
This is a fair question. In the article cited by Terpos et al [37], they refer to their analyses as “ad hoc.” However, they may be most consistent with post-hoc analyses as the article was published three years after the original study [36]. We have left it as is to stay consistent with the authors’ description of their research but are open to changing the wording to “post hoc” if the journal feels this is more appropriate.
References: I am not a fan of citing UptoDate; it may change over time and have some element of opinion. I would prefer citing references found within UptoDate, or a package insert that includes its year of publication.
Thank you for pointing this out. We agree, and UpToDate was used only to cite pharmacologic parameters. We have updated our references to include FDA and manufacturer labels instead.
Reviewer 3 Report
Comments and Suggestions for Authors
Overall this is a well written and detailed review
The data on improved PFS with denosumab vs zometa in a particular subgroup should be played down in the abstract -the quality of evidence is not sufficient to say one agent is better than the other. Sure agree its intriguing but would only be confirmed in a proper large randomised prospective study
I think there are two major omissions in this review and I think would hopefully enhance the review
- Smouldering myeloma – a reader will want to know what to do here and there are three studies that are not mentioned at all. I think most guidelines do not recommend bisphopshonates but I am not sure why – worth including this given there is an increasing move to treat SMM sooner these days
UK SMM guidelines refer to three RCT of bisphosphonates in SMM where there was no effect on time to progression to myeloma but significant effects on SRE
Musto 2003. 45 pamidronate 45 observation; skeletal related events at progression were lower in the pamidronate group, 40% vs 81.8% p<0.001
Musto 2008. 81 zoledronate 82 observation: skeletal related events at progression were lower in the zoledronate group, 55.5% vs 78% p<0.041
D’Arena 2011. 89 pamidronate 88 observation; skeletal related events at progression were lower in the pamidronate group, 39.2% vs 72.7% p<0.009
- I think this is obvious but bone mass/health is critically linked to physical activity and muscle health so needs to be at least some comment on the importance of physiotherapy and encouraging some level of activity and any other activity to reduce frailty especially given the median age of myeloma is around 70 yo
The sentence could perhaps be improved - Patients should receive adequate calcium and vitamin D supplementation while supportive procedures such as cement augmentation, radiation, and surgery can also help treat compression fractures, uncontrolled pain, cord compression, and pathologic fractures. For example - Patient should receive adequate calcium and vitamin D supplementation. Cement augmentation, radiation, surgery and bracing can be considered for spinal disease although there is little evidence in this area and orthopaedic input is required for fractures.
Line 30 - An estimated 35,730 30 new cases were diagnosed in 2023 … In USA I presume? Please clarify
Line 56 Multiple myeloma cells also de-grade osteoprotegerin (OPG), a soluble decoy receptor for RANKL that antagonizes osteoclastogenesis, after internalizing it via binding to heparan sulfate proteoglycans (HSPGs) on the myeloma cell surface. Myeloma cells also bind and deactivate OPG via syndecan-1, a transmembrane proteoglycan with heparan sulfate side chains, on the cell surface. [8]
Syndecan is a HSPG so these two sentences are confusing. Why not just say
Multiple myeloma cells also degrade osteoprotegerin (OPG), a soluble decoy receptor for RANKL that antagonizes osteoclastogenesis, after internalizing it via binding to heparan sulfate proteoglycans (HSPGs) such as syndecan-1 (CD138) on the myeloma cell surface. [8]
Line 59 – worth saying syndecan-1 is CD138 as the reader may more likely to be familiar with that.
Line 172 – my theory is that compliance with oral sodium clodronate may be poor – it has some toxicity especially gut. This may explain some of the inferiority seen in the Morgan et al trial where it was compared to zoledronate
Page 317 we do not feel that it refutes the potential PFS advantage for denosumab found by 317 Raje and Terpos. -better english here ?
Author Response
The data on improved PFS with denosumab vs zometa in a particular subgroup should be played down in the abstract -the quality of evidence is not sufficient to say one agent is better than the other. Sure agree its intriguing but would only be confirmed in a proper large randomised prospective study
Thank you for this comment. We agree that we cannot definitively say that one agent is strongly preferred over the other. We have adjusted the manuscript to state that denosumab is likely preferred in patients with plans to undergo ASCT and receiving proteosome inhibitor treatment. These changes were made to the abstract, recommendations for the management of bone disease in NDMM section, and conclusion. Figure 2 was also adjusted to reflect these changes.
I think there are two major omissions in this review and I think would hopefully enhance the review
- Smouldering myeloma – a reader will want to know what to do here and there are three studies that are not mentioned at all. I think most guidelines do not recommend bisphopshonates but I am not sure why – worth including this given there is an increasing move to treat SMM sooner these days
UK SMM guidelines refer to three RCT of bisphosphonates in SMM where there was no effect on time to progression to myeloma but significant effects on SRE
Musto 2003. 45 pamidronate 45 observation; skeletal related events at progression were lower in the pamidronate group, 40% vs 81.8% p<0.001
Musto 2008. 81 zoledronate 82 observation: skeletal related events at progression were lower in the zoledronate group, 55.5% vs 78% p<0.041
D’Arena 2011. 89 pamidronate 88 observation; skeletal related events at progression were lower in the pamidronate group, 39.2% vs 72.7% p<0.009
Thank you for this suggestion. While it is an excellent topic to discuss, we feel that preventing skeletal related events in smoldering myeloma is out of the scope of our review on preventing skeletal related events in newly diagnosed multiple myeloma. We thus have not added additional discussion on this topic. If the journal feels that it would be beneficial to add this discussion to our review, we can do so if given extra time to make these additions but again we feel this is a separate topic outside the scope of this review.
I think this is obvious but bone mass/health is critically linked to physical activity and muscle health so needs to be at least some comment on the importance of physiotherapy and encouraging some level of activity and any other activity to reduce frailty especially given the median age of myeloma is around 70 yo
Thank you for pointing this out. We have added a statement discussing a study by Larsen et al (2024) examining structured exercise programs in MM patients. While it is a negative study, it showed that structured exercise is safe in MM patients. We note that exercise is likely beneficial to MM patients even though we do not have specific data measuring bone turnover markers in MM patients. Lines 477-480.
The sentence could perhaps be improved - Patients should receive adequate calcium and vitamin D supplementation while supportive procedures such as cement augmentation, radiation, and surgery can also help treat compression fractures, uncontrolled pain, cord compression, and pathologic fractures. For example - Patient should receive adequate calcium and vitamin D supplementation. Cement augmentation, radiation, surgery and bracing can be considered for spinal disease although there is little evidence in this area and orthopaedic input is required for fractures.
Thank you for this comment. We have adjusted the sentence structure in the abstract and Recommendations for the Management of Bone Disease in NDMM section to reflect the recommended clearer language.
Line 30 - An estimated 35,730 30 new cases were diagnosed in 2023 … In USA I presume? Please clarify
Thank you for this comment. The manuscript has been adjusted to reflect 2025 numbers and now notes that it refers to patients in the United States.
Line 56 Multiple myeloma cells also de-grade osteoprotegerin (OPG), a soluble decoy receptor for RANKL that antagonizes osteoclastogenesis, after internalizing it via binding to heparan sulfate proteoglycans (HSPGs) on the myeloma cell surface. Myeloma cells also bind and deactivate OPG via syndecan-1, a transmembrane proteoglycan with heparan sulfate side chains, on the cell surface. [8]
Syndecan is a HSPG so these two sentences are confusing. Why not just say
Multiple myeloma cells also degrade osteoprotegerin (OPG), a soluble decoy receptor for RANKL that antagonizes osteoclastogenesis, after internalizing it via binding to heparan sulfate proteoglycans (HSPGs) such as syndecan-1 (CD138) on the myeloma cell surface. [8]
Line 59 – worth saying syndecan-1 is CD138 as the reader may more likely to be familiar with that.
Thank you for pointing this out. We agree that the language can be clarified. The sentence was adjusted to reflect the recommend changes. Lines 71-74
Line 172 – my theory is that compliance with oral sodium clodronate may be poor – it has some toxicity especially gut. This may explain some of the inferiority seen in the Morgan et al trial where it was compared to zoledronate
Thank you for this comment. On further review of the Myeloma IX trial (Morgan et al 2010), the treatment discontinuation rates prior to disease progression reported between both groups are similar (24% for ZA and 19% for clodronate). Our manuscript has been adjusted to note these discontinuation rates as well.
Page 317 we do not feel that it refutes the potential PFS advantage for denosumab found by 317 Raje and Terpos. -better english here ?
Thank you for pointing this out. The sentence structure has been adjusted for clarity. (Lines 463-464)
Round 2
Reviewer 1 Report
Comments and Suggestions for Authors
The manuscript presents a comprehensive review of skeletal-related event (SRE) prevention in newly diagnosed multiple myeloma (NDMM), including discussion of the underlying pathophysiology, therapeutic options (bisphosphonates and RANKL inhibitors), and supportive care measures. While the topic is relevant and timely, several important issues should be addressed before the manuscript can be considered for publication.
- Clarification of Ambiguous and Grammatically Incorrect Sentence
Original:
“Both zoledronic acid (ZA) and denosumab are acceptable treatment options and have been comparedare similar in terms of safety and efficacy; however. In patients with plans to undergo autologous stem cell transplant, denosumab is likely preferred over ZA given thethat a progression-free survival benefit was observed when combined with proteosome inhibitors…”
Suggested Revision:
“Both zoledronic acid (ZA) and denosumab are acceptable treatment options with comparable safety and efficacy profiles. However, in patients who are candidates for autologous stem cell transplant (ASCT), denosumab may be preferred over ZA due to a progression-free survival (PFS) benefit observed in post hoc analyses when used in combination with proteasome inhibitor-based regimens.”
- Overstatement of Data Interpretation
Original:
“We would interpret this data as providing strong support to the idea that denosumab is non-inferior to ZA, but. However, we do not feel that it refutes the data refute denosumab’s potential PFS advantage for denosumab found by Raje and Terposin transplant eligible patients receiving proteosome inhibitors.”
Suggested Revision:
“We interpret these findings as supporting the non-inferiority of denosumab relative to ZA. Although the retrospective cohort by Mohyuddin et al did not demonstrate a significant PFS or OS benefit, the study population differed substantially from those included in prior prospective trials. Therefore, while the data do not confirm a PFS benefit, they also do not refute the possibility of such benefit in appropriately selected transplant-eligible patients receiving proteasome inhibitors.”
- Clarify Statistical Significance vs. Trends
In several instances, the manuscript appears to interpret non-significant trends as definitive outcomes. For example, the statement regarding denosumab "trending toward superiority" should be clearly distinguished from statistically significant findings.
Suggested Addition:
“Where applicable, please clarify whether observed differences are statistically significant or represent trends that require cautious interpretation.”
Comments on the Quality of English LanguageThe proficiency in the English language can be enhanced.
Author Response
- Clarification of Ambiguous and Grammatically Incorrect Sentence
Original:
“Both zoledronic acid (ZA) and denosumab are acceptable treatment options and have been comparedare similar in terms of safety and efficacy; however. In patients with plans to undergo autologous stem cell transplant, denosumab is likely preferred over ZA given thethat a progression-free survival benefit was observed when combined with proteosome inhibitors…”
Suggested Revision:
“Both zoledronic acid (ZA) and denosumab are acceptable treatment options with comparable safety and efficacy profiles. However, in patients who are candidates for autologous stem cell transplant (ASCT), denosumab may be preferred over ZA due to a progression-free survival (PFS) benefit observed in post hoc analyses when used in combination with proteasome inhibitor-based regimens.”
Thank you for this suggestion. We agree this language conveys the information presented more effectively. We have updated the manuscript with the recommended sentence structure in our manuscript. Lines 12-16.
-----------------------------------------------------------------------------------------------------------
- Overstatement of Data Interpretation
Original:
“We would interpret this data as providing strong support to the idea that denosumab is non-inferior to ZA, but. However, we do not feel that it refutes the data refute denosumab’s potential PFS advantage for denosumab found by Raje and Terposin transplant eligible patients receiving proteosome inhibitors.”
Suggested Revision:
“We interpret these findings as supporting the non-inferiority of denosumab relative to ZA. Although the retrospective cohort by Mohyuddin et al did not demonstrate a significant PFS or OS benefit, the study population differed substantially from those included in prior prospective trials. Therefore, while the data do not confirm a PFS benefit, they also do not refute the possibility of such benefit in appropriately selected transplant-eligible patients receiving proteasome inhibitors.”
Thank you for this comment. We agree that this language communicates the intended interpretation of the data more clearly. We have updated the manuscript with the recommended revision in our manuscript. Lines 343-348.
-----------------------------------------------------------------------------------------------------------
- Clarify Statistical Significance vs. Trends
In several instances, the manuscript appears to interpret non-significant trends as definitive outcomes. For example, the statement regarding denosumab "trending toward superiority" should be clearly distinguished from statistically significant findings.
Suggested Addition:
“Where applicable, please clarify whether observed differences are statistically significant or represent trends that require cautious interpretation.”
Thank you for pointing this out. We agree that the language about denosumab trending towards superiority regarding time to first SRE in Henry et al (2011) should be more clearly distinguished from the statistically significant non-inferiority analysis. We have updated the manuscript on lines 254-256.
Additionally, we have added details to our interpretation of the landmark analysis at 15 months done by Raje et al (2018) that shows denosumab was superior compared to ZA in terms of time to first SRE. We have more clearly noted that this was a post hoc landmark analysis. Lines 296-299 and 448-449.
Lastly, we have removed a statement that ZA increased median PFS by 2 months compared to clodronate to avoid confusion as this was not statistically significant (p=0.07). Lines 186-187.